

# Effects of sandfish (*Holothuria scabra*) removal on shallow-water sediments in Fiji

Steven Lee[1,2], Amanda K. Ford[1,2], Sangeeta Mangubhai[3], Christian Wild[2] and Sebastian C.A. Ferse[1,2]

[1] Leibniz Centre for Tropical Marine Research (ZMT), Bremen, Germany
[2] Faculty of Biology and Chemistry (FB2), University of Bremen, Bremen, Germany
[3] Fiji Country Program, Wildlife Conservation Society, Suva, Fiji

## ABSTRACT

Sea cucumbers play an important role in the recycling and remineralization of organic matter (OM) in reef sands through feeding, excretion, and bioturbation processes. Growing demand from Asian markets has driven the overexploitation of these animals globally. The implications of sea cucumber fisheries for shallow coastal ecosystems and their management remain poorly understood. To address this knowledge gap, the current study manipulated densities of *Holothuria scabra* within enclosures on a reef flat in Fiji, between August 2015 and February 2016, to study the effects of sea cucumber removal on sedimentary function as a biocatalytic filter system. Three treatments were investigated: (i) *high* density (350 g m$^{-2}$ wet weight; *ca.* 15 individuals); (ii) *natural* density (60 g m$^{-2}$; *ca.* 3 individuals); and (iii) *exclusion* (0 g m$^{-2}$). Quantity of sediment reworked through ingestion by *H. scabra*, grain size distribution, O$_2$ penetration depth, and sedimentary oxygen consumption (SOC) were quantified within each treatment. Findings revealed that the natural population of *H. scabra* at the study site can rework *ca.* 10,590 kg dry sediment 1,000 m$^{-2}$ year$^{-1}$; more than twice the turnover rate recorded for *H. atra* and *Stichopus chloronotus*. There was a shift towards finer fraction grains in the *high* treatment. In the *exclusion* treatment, the O$_2$ penetration depth decreased by 63% following a 6 °C increase in water temperature over the course of two months, while in the *high* treatment no such change was observed. SOC rates increased *ca.* two-fold in the *exclusion* treatment within the first month, and were consistently higher than in the *high* treatment. These results suggest that the removal of sea cucumbers can reduce the capacity of sediments to buffer OM pulses, impeding the function and productivity of shallow coastal ecosystems.

Corresponding author
Steven Lee, steven.d.a.lee@gmail.com

## INTRODUCTION

Marine coastal ecosystems are among the most productive and diverse on earth (*Poore & Wilson, 1993*). Coastal communities are reliant on the various ecosystem functions and services these areas provide for their livelihoods (*Conservation International, 2008*). Recent studies have identified significant ecological impacts when benthic marine organisms

were removed from these ecosystems (*Lohrer, Thrush & Gibbs, 2004*; *Solan et al., 2004*). A loss in bioturbation (the biogenic mixing of sediment) has been demonstrated following the reduction in abundance and diversity of marine benthic fauna (*Solan et al., 2004*). This loss is of particular concern as bioturbation has a substantial influence on the rate of organic matter (OM) decomposition and nutrient recycling (*Meysman et al., 2006*). Efficient processing of OM and nutrients in marine coastal ecosystems is crucial to the health and productivity of these ecosystems, and as such is vital to the coastal communities dependent on them.

The majority of OM in marine coastal ecosystems is trapped in permeable reef sands and degraded by the infaunal community, particularly high densities of sand-associated microbes (*Wild et al., 2004*). Due to the high surface area of permeable carbonate reef sands and advective flow into and within the sediment, these sandy sediments promote efficient degradation of OM (*Wild et al., 2004*; *Wild, Laforsch & Huettel, 2006*). Such sandy sediments have consequently been referred to as a biocatalytic filter system, promoting critical recycling processes (*Wild, Tollrian & Huettel, 2004*; *Huettel, Wild & Gonelli, 2006*). Benthic-pelagic coupling implies that changes in the OM concentration of overlying water are integrated and reflected in sediments, and furthermore implies that changes in sediment OM composition affect the overlying water quality (*Wild, Tollrian & Huettel, 2004*). Thus, the efficient functioning of this biocatalytic filter system provides coastal ecosystems—which are increasingly stressed by OM enrichment from anthropogenic inputs (*Halpern et al., 2008*; *Rabalais et al., 2009*)—with buffering capacity. Without the capacity of the system to buffer increasing OM loads, the health of coastal ecosystems could be compromised, resulting for example in shifts from hard coral- to algal-dominated reefs (*Fabricius, 2005*; *D'Angelo & Wiedenmann, 2014*).

Coral reefs are a critically important component of coastal ecosystems for the ecosystem services and goods they provide, such as fisheries, coastal protection, and nutrient cycling (*Moberg & Folke, 1999*). Inshore reefs and their associated ecosystems receive a considerable amount of OM from rivers and marine sources. The quantity of OM that reaches inshore reefs increases following heavy rainfall and flooding events (*Briand et al., 2015*), delivering a 'pulse' of OM that places further stress on the ecosystem. The efficiency of the system to absorb this stress and act as a biocatalytic filter is dependent on a number of factors including temperature, OM supply, light, water currents, and bioturbation (*Kristensen, 2000*). Bioturbation is particularly important for the efficiency of this filter system (*Kristensen, 2000*) as it increases the surface area of the sediment, breaks down geochemical gradients, and maximizes advective flow into the sediment (*Thibodeaux & Boyle, 1987*; *Kristensen, 2000*; *Lohrer, Thrush & Gibbs, 2004*; *Meysman, Middelburg & Heip, 2006*).

Deposit feeding sea cucumbers mainly of the order Holothuriida potentially play a significant role in enhancing OM mineralization through bioturbation as well as through feeding and excretion (*Uthicke, 1999*; *Uthicke, 2001a*; *Uthicke, 2001b*; *Purcell, 2004*; *Wolkenhauer et al., 2010*; *MacTavish et al., 2012*). Whilst sea cucumbers were once a ubiquitous component of many coastal marine benthic communities, growing demand from Asian markets for their dried form—known as bêche-de-mer—has driven their overexploitation, leading to population collapse and local extinctions throughout the
Indo-Pacific (e.g., *Toral-Granda, Lovatelli &, Vasconcellos, 2008* ; *Friedman et al., 2009*; *Friedman et al., 2011*; *Purcell et al., 2013*). Sandfish, *Holothuria scabra* (*Jaeger, 1833*), is one of the highest valued species in the bêche-de-mer trade (*Purcell, 2014*). Individuals of this species are easily accessible to fishers as they generally inhabit low energy environments behind fringing reefs or within protected bays and shores (*Hamel et al., 2001*), rendering the species particularly vulnerable to overexploitation. *H. scabra* exhibits a natural diurnal burying cycle and ingests a considerable amount of sediment during feeding, thus playing a key role in bioturbation (*Mercier, Battaglene & Hamel, 1999*; *Purcell, 2004*). Historically this species was found in densities of up to 2 individuals m$^{-2}$ (*Hamel et al., 2001*), implying that its bioturbation impact on inshore reef habitats is likely to have been substantial.

Inshore reef ecosystems are very dynamic environments in which conditions exhibit a large variation of temperature, salinity, turbidity, and wave energy, all of which influence sediment function and sea cucumber behavior (*Mercier, Battaglene & Hamel, 1999*; *Mercier, Battaglene & Hamel, 2000*). Initial studies investigating the ecological roles of sea cucumbers and sediment function have mainly been conducted *ex situ* or using mesocosms (e.g., *Uthicke, 1999*). Given the nature of inshore reef ecosystems, *in situ* studies are required to obtain more realistic results (see *Wolkenhauer et al., 2010*; *Namukose et al., 2016*).

Sedimentary O$_2$ consumption (SOC) and sediment O$_2$ penetration depth (OPD) provide a proxy for the function of sediment as a biocatalytic filter system. SOC reflects the respiration of the entire sedimentary community contained in a core and integrates OM deposited from the overlying water column (*Nickell et al., 2003*). O$_2$ penetration depth (OPD) determines the REDOX reactions occurring within the sediment, and thus the volume of sediment effectively participating in aerobic OM decomposition (*Glud, 2008*). Furthermore, sea cucumber feeding can alter grain size distribution through dissolution of calcium carbonate via acidity and potentially abrasion processes in their gut (*Hammond, 1981*; *Schneider et al., 2011*). As the composition of sediment grain size influences sediment O$_2$ dynamics (*Urumović & Urumović, 2014*), changes in grain size distribution caused by sea cucumbers may account for changes in SOC and OPD.

The current study aimed to understand how the removal of sea cucumbers affects sedimentary function through *in situ* experimental manipulations of *H. scabra* densities, thus providing information on the poorly understood ecological implications of sea cucumber fisheries. We hypothesized that high densities of *H. scabra* would facilitate the efficient degradation of OM. Thus, in areas with *H. scabra,* SOC rates were expected to decrease or remain relatively lower, and OPD was expected to increase compared to those areas devoid of *H. scabra*. Finally, sediment turnover—the amount of sediment ingested by *H. scabra*—was quantified as a proxy for the bioturbation potential of the animal.

## METHODS

### Study site

The experiment was conducted between August 2015 and February 2016 on an extensive reef flat in front of Natuvu village, Wailevu East District, Vanua Levu, Fiji (16°44.940′S 179°9.280′E) (Fig. 1). The reef flat was part of the locally-managed marine area (LMMA)
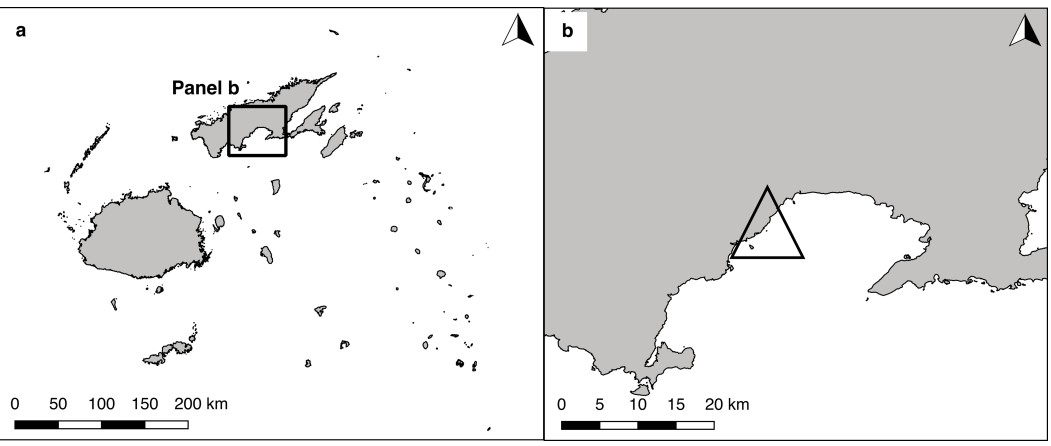

**Figure 1** **Map of the study site.** Location of Wailevu East District, Vanua Levu, Fiji (A), and the location of the study site (Natuvu) as indicated by a triangle (B).

of the local villages. The entire reef flat covered an area of roughly 1.2 km², and contained a *tabu* area (periodically harvested closure). This *tabu* area supported a relatively high density of *H. scabra* compared to the wider region, which was closer to regional reference values (*Pakoa et al., 2013*). The remoteness of the study site and lack of electricity and infrastructure unfortunately prohibited reliable OM measurements.

A pilot survey of the site using 100 m × 2 m belt transects was conducted various times throughout the day and night, and during various tides in order to optimize estimates of the population density, and burying and feeding behavior of *H. scabra* throughout a 24 h cycle.

## Enclosure design and construction

Sixteen square plots were demarcated on the reef flat, comprising four treatments: three types of enclosures and one control ($n = 4$ treatment$^{-1}$). Treatments included (i) high density (*high*) (*ca.* 350 g m$^{-2}$—based on the *in situ* carrying capacity for *H. scabra*) (*Namukose et al., 2016*; *Lavitra, Rasolofonirina & Eeckhaut, 2010*; *Purcell & Simutoga, 2008*; *Battaglene, Evizel & Ramofafia, 1999*), (ii) exclusion (*exclusion*) (0 g m$^{-2}$—simulating overexploitation), (iii) natural density (*natural*) (*ca.* 60 g m$^{-2}$—determined by a pilot survey at the study site), and (iv) control plots (*uncaged control*, plots without mesh). Comparing between natural density cages and uncaged control treatments, which had the same density of sea cucumbers, allowed assessing the effect of the presence of mesh cages. Enclosures were stocked with *H. scabra* of length *ca.* 15 cm, which equates to *ca.* 200 g ind$^{-1}$, as this was the most common length of *H. scabra* in the general area where enclosures were situated. Stocking densities were monitored six days per week and enclosures restocked if necessary. *Ca.* 15 *H. scabra* were maintained in high-density enclosures and approximately three in natural-density enclosures. All animals used to stock enclosures were collected from beyond 500 m outside the study site.

The lengths and weights of *H. scabra* were recorded following similar methods used by *Seeto (1994)* and *Al-Rashdi, Claereboudt & Al-Busaidi (2007)*. *H. scabra* individuals were removed from the water and allowed to initially contract and expel water, which
happened almost immediately following handling (<3 s). The animal's length was recorded from anus to mouth by placing a ruler along its ventral surface. All measurements were recorded to the nearest centimeter. The specimens were then placed into containers filled with water from the site. Water in the containers was continuously exchanged to reduce stress on the animals. All *H. scabra* of the same length were kept together in containers. Following each collection, the animals were taken ashore and weighed on a digital scale ($\pm 0.02$ g) to the nearest gram. Time from initial capture to weighing was no longer than 1 h. Individuals were allowed to contract and expel water before being weighed, as described above. Following weighing, all sea cucumbers were released back to the site.

Enclosures (3 × 3 m) were designed according to recommendations by *Miller & Gaylord (2007)* in order to minimise cage effects. Each enclosure was constructed using eight 1.5 m pieces of 16 mm steel re-bar driven vertically into the sediment, with walls comprised of high-density polyethylene diamond mesh with a 40 mm aperture. Re-bars were woven through the mesh and driven *ca*. 20 cm into the sediment, leaving *ca*. 80 cm of mesh exposed. Enclosure construction was completed on the second week of September following initial measurements ($T = 0$).

Enclosures were monitored for any signs of damage, disturbance, or need for restocking once a day—at low tide—six days per week (the community did not allow any work on Sunday for religious reasons). All animals used to restock enclosures were collected from the immediate area around each respective enclosure. The weather was monitored throughout the course of the study using national and regional weather reports (http://www.met.gov.fj), as well as anecdotal reports from villagers at the study site.

Primary sediment sampling was conducted during the first week of each month. A plot from each treatment (i.e., *high*, *exclusion*, *natural*, and *uncaged control*) was randomly selected for sediment sampling each day. Sampling generally took place between 0700 h and 0900 h, which coincided with the rising tide, as dissolved $O_2$ levels during this time were at their optimum for SOC measurements.

## Grain size distribution

Sediment samples ($n = 3$ treatment$^{-1}$) were collected using a corer made from a clear 50 ml syringe (core depth = *ca*. 30 mm, diameter = *ca*. 30 mm). Sediment cores for each plot were transferred into individual airtight containers. Sediment cores were washed out of their containers onto a dish using freshwater, then placed in a dry oven at 70 °C for *ca*. 12 h. Dry sediment samples were weighed and transferred to the top of a sieve column ($\geq 2,000$, 1,000, 500, 250, 125, $\leq 125$ μm) attached to a sieve shaker. The sieve column was shaken for seven minutes, and the contents of each sieve then weighed using a ViBRA analytical balance ($\pm 0.02$ g). If salt in the sediment samples were to influence results, it is likely the smallest grain size fractions would have been affected.

## Oxygen penetration depth

Sediment cores from the study site exhibited a clear anoxic layer characterized by consistently dark sediment, which indicates sulphate reduction to hydrogen sulphide by sulphate-reducing bacteria (*Castro & Huber, 2012*). Therefore, it was possible to determine

OPD visually by measuring the distance from the top of the core to the black layer within sediment cores ($n = 3$ treatment$^{-1}$, to a depth $\geq 3$ cm) collected in clear 50 ml syringes, following methods adapted from *Nilsson & Rosenberg (2000)*. OPD was only measured from $T = 2$ onwards, due to time constraints during $T = 0$ and the initial method trialed during $T = 1$ being unsuitable.

## Sedimentary oxygen consumption

$O_2$ consumption of sediments is driven by respiration of benthic organisms, microbially-mediated oxidation of OM, and reduced inorganic metabolites (*Kristensen, 2000*), therefore SOC can indicate OM input and degradation at a local level. *In situ* determination of SOC was based on methods used by *Ford et al. (2017)*. Sediment cores (10 cm$^3$) were collected from enclosures using cut 50 ml syringes and transferred immediately to 160 ml glass incubation chambers. Chambers were subsequently filled with undisturbed water from the overlying water column ($n = 4$ treatments enclosure$^{-1}$). $O_2$ concentration, salinity, and temperature were measured using an $O_2$ optode sensor and conductivity probe (MultiLine® IDS 3430; WTW GmbH, Weilheim, Germany; accuracy: ±0.5% of measured value). $O_2$ saturation was consistently between 70–120% at initial measurements. Chambers were sealed airtight before being placed into opaque bags and then placed in an icebox filled with water from the site to maintain a consistent temperature and to ensure light was excluded. After *ca.* 1 h incubation, chambers were collected and $O_2$ concentration was re-measured. The exact duration (minutes) of each incubation was recorded. Salinity and seawater temperature within the chambers were re-measured to ensure consistency throughout measurements. Controls ($n = 3$ enclosure$^{-1}$) contained only undisturbed water from our site, allowing us to account for microbial activity in the overlying water column. $O_2$ consumption in control chambers was averaged and subtracted from sediment SOC chamber rates. SOC values were calculated into mmol $O_2$ m$^{-2}$ sediment d$^{-1}$ after accounting for incubation time, vial volume and control measurements.

## Sediment turnover

Daily sediment turnover (proxy for bioturbation) by *H. scabra* was calculated by assuming the quantity of defecated sediment was equal to the quantity of ingested sediment (*Uthicke, 1999*). *H. scabra* between lengths of 8–16 cm ($n = 27$, mean length = 13 cm ± 0.22 SE; mean mass = 145 g ± 6.27 SE) were selected for this parameter as they represent the most abundant size class across the entire reef flat and were found in the three most common habitats outside the enclosures: sand ($n = 17$), *Halodule* spp. bed ($n = 6$), and *Syringodium* spp. bed ($n = 4$) (*Lee, 2016*). Data were collected at flood and ebb tide during both night and day over the course of two weeks in January 2016 (0800 h–2100 h, and random checks for feeding behavior between 0100 h–0500 h). The methodology for this experiment was adapted from a similar study by *Uthicke (1999)*.

Sediment defecated was quantified from trails of faeces behind *H. scabra*. Markers were driven vertically into the sediment at a standardised distance (*ca.* 1 cm) behind the posterior end of the animal. Approximately every hour (for 4 h) the animal was relocated, and an additional marker placed at its posterior end. The animal's length and weight were

recorded, and time was noted each time a marker was placed. Six pellets per animal were collected in individual vials during the four hours that the animal was monitored, and the number of pellets between each marker was counted. Sediment pellets were placed in a dry oven at 70 °C for approximately 12 h before weighing on an analytical balance (±0.02 g). Bulk density of sediment at the study site was calculated by comparing the volume of sediment cores to the dry weight of sediment contained within.

### Data analysis

Grain size analysis, textural classifications and distribution of sediments were carried out in R using the *gran.stats* function (*rysgran* package; *Gilbert, Camargo & Sandrini-Neto, 2014*) based on methods and verbal classifications by *Folk & Ward (1957)*. Differences in skewness, kurtosis, and median grain size were tested using the *aov* function in R (built-in function; *R Core Team, 2016*). The effect of treatment and time on SOC and OPD data were analysed using repeated-measures analyses of variances using the *aov* function in R (built-in function; *R Core Team, 2016*), followed by post-hoc students *t*-tests or Mann Whitney–Wilcox *U* test (*t.test* and *pairwise.wilcox.test*—built-in function; *R Core Team, 2016*) as appropriate. All multiple comparison post-hoc tests were performed with Bonferroni correction. Paired tests were used to compare a treatment over time, whereas non-paired tests were applied to compare among treatments for a single sample date. Data were tested for normality using a Shapiro–Wilk test and a Quantile-Quantile plot of the residuals, and for homogeneity of variances using a Levene's test (*leveneTest*—*car* package; *Fox & Weisberg, 2011*). SOC and OPD data were transformed (log10) using R in order to meet assumptions of normality. If these data did not meet the assumptions then the non-parametric Mann Whitney–Wilcox *U* test (*pairwise.wilcox.test*—built-in function; *R Core Team, 2016*) was used. *Uncaged control* and *natural* treatments were tested against each other for significant differences at each sampling date; if no significant differences were found then *high* treatment was compared to *exclusion* treatment, and *natural* was compared to *exclusion* and *high* treatments.

## RESULTS

As indicated above, primary sediment sampling was conducted during the first week of each month. Sampling began in September 2015 ($T = 0$) and continued each month until February 2016 ($T = 5$). As enclosures were checked daily and restocked if necessary to maintain stocking densities, the effect of loss of *H. scabra* between restocking was deemed minimal; overall, restocking was in the range of two to three animals in the *high* treatment enclosures and one to two animals in cages of the *natural* treatment. *H. scabra* would occasionally have to be removed from exclusion treatment enclosures.

### Environmental parameters

Water temperature as quantified during initial SOC measurements showed a steady increase between September ($T = 0$; 23.8 °C ± 0.23 mean ± SE) and December 2015 ($T = 3$; 26.6 °C ± 0.19), followed by a severe increase between December 2015 and

February 2016 ($T = 5$; 31.2 °C $\pm$ 0.07) (Fig. 2A). Heavy rain 10 days prior to sampling in November ($T = 2$; Fig. 2C) caused coastal flooding, and a tropical depression (Tropical Disturbance 05F; *Fiji Meteorological Service, 2015*) five days prior to sampling in January ($T = 4$; Fig. 2C) caused coastal flooding and storm surge at the site.

### Grain size distribution

Throughout the course of the study within all treatments, mean skewness was between 0.01 and 0.07, i.e., near symmetrical, and mean kurtosis was between 0.88 and 0.95, i.e., approximately mesokurtic. Therefore, grain size distribution was approximately normal. Prior to the onset of the experiment ($T = 0$) there were neither significant differences in grain size fractions between controls (*uncaged control* vs. *natural treatment*) nor experimental treatments (*high* vs. *exclusion*) (ANOVA; $p > 0.05$). The *uncaged control* and *natural* treatments remained consistent between $T = 0$ and $T = 3$ (ANOVA; $p = {>}0.05$). Over the same time period the *high* treatment did exhibit two trends, though they are contradictory and neither was significant (ANOVA; $p > 0.05$). First, there was an increase in median grain size from 0.91 $\pm$ 0.09 phi (mean $\pm$ SE) to 1.07 $\pm$ 0.06 phi (mean $\pm$ SE), and second, skewness tended to decrease from 0.04 $\pm$ 0.03 (mean $\pm$ SE) at $T = 0$ to 0.01 $\pm$ 0.03 (mean $\pm$ SE) at $T = 3$. Despite storm surge affecting the study site five days prior to $T = 4$, there were no significant differences between treatments at $T = 4$ and $T = 5$ (ANOVA; $p > 0.05$). Changes to grain size distribution between $T = 3$ and $T = 4$ were not considered, as the storm surge between these sampling events had visibly shifted sediments.

### Oxygen penetration depth

The interaction of treatment and time had a significant effect on OPD (Repeated measures ANOVA; $F(9,116)$, $p = 0.02$). OPD measurements began at $T = 2$ of the experiment. Initial ($T = 2$) OPD were similar between treatments and controls (Mann–Whitney $U$-test; $p > 0.05$). By $T = 3$ the only significant change was an increased OPD in *high* treatment enclosures (Mann–Whitney $U$-test; $p = 0.03$). The *exclusion* treatment remained consistent (Mann–Whitney $U$-test; $p = 0.37$) between $T = 2$ and $T = 3$. A distinctly different pattern emerged at $T = 5$. Whilst *high* treatment enclosures returned to their earlier state ($T = 5$ vs. $T = 2$; $U$-test, $p = 1$), OPD in sediment from *exclusion* treatment enclosures decreased from 32 mm $\pm$ 3 at $T = 3$ to 12 mm $\pm$ 2 (mean $\pm$ SE) at $T = 5$, exhibiting a 63% reduction and resulting in a highly significant difference between treatments (Mann–Whitney $U$-test; $p < 0.01$). While *natural* and *uncaged control* varied over the course of the entire study (Fig. 2B), there were no significant differences between *natural* and *uncaged control* at any individual sampling date.

### Sedimentary oxygen consumption

The interaction of treatment and time had a significant effect on SOC (Repeated measures ANOVA; $F(15,360)$, $p < 0.01$). Between $T = 0$ and $T = 1$, SOC in *exclusion* treatment areas initially increased almost two-fold from 43.03 mmol $O_2$ m$^{-2}$ day$^{-1}$ $\pm$ 4.59 to 75.96 mmol $O_2$ m$^{-2}$ day$^{-1}$ $\pm$ 4.69 (Mann–Whitney $U$-test; $p < 0.01$, Fig. 2C). Over the same time period, *high* treatment SOC rates remained consistent (Mann–Whitney $U$-test, $p = 0.74$). Sediment in *exclusion* treatment areas had a noticeably higher SOC than *high*

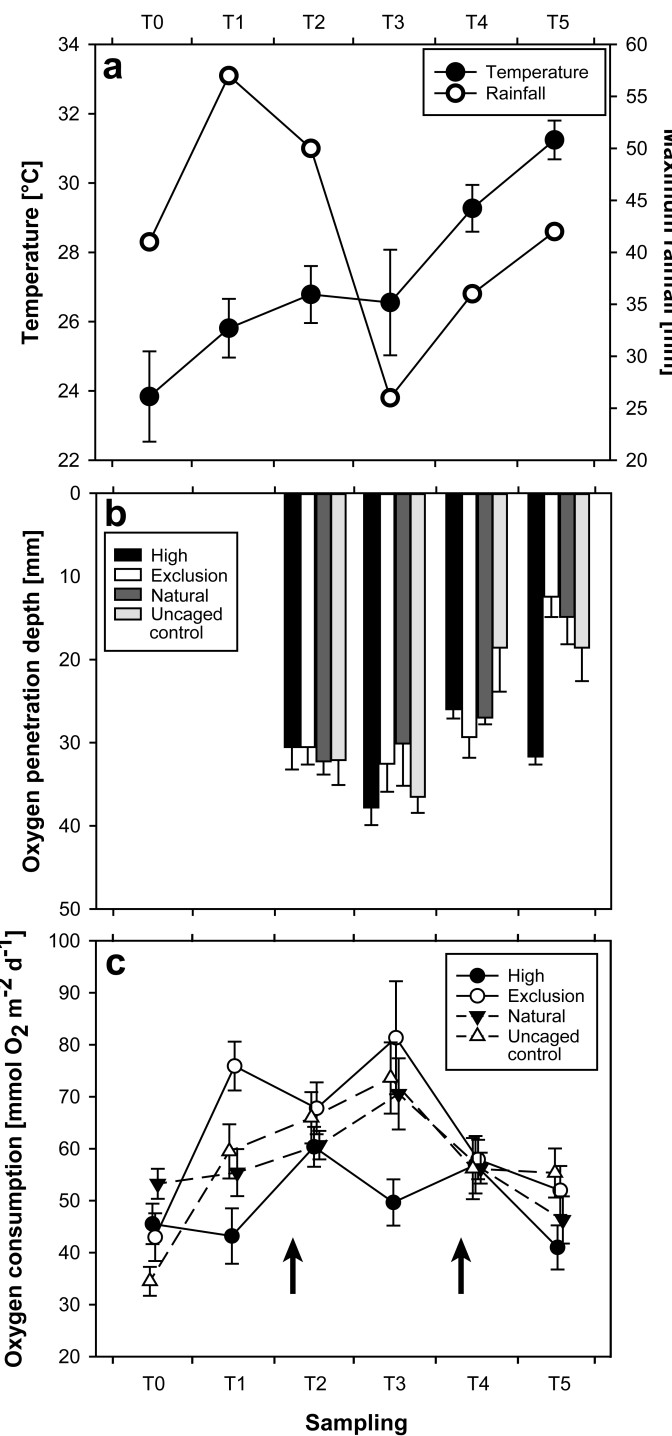

**Figure 2** **Plots of results for temperature and rainfall (A), oxygen penetration depth (B), and sediment oxygen consumption (C), at each sampling time.** Arrows in 2C indicate the timing of two particularly strong rainfall events.

**Table 1 Amount of sediment bioturbated by a population of *Holothuria scabra*.** Average *H. scabra* densities are taken from a pilot survey (S Lee, 2015, unpublished data), quantity of sediment defecated is assumed equal to that consumed, and sea cucumbers are assumed to consume only the upper 5 mm of sediment (*Uthicke, 1999*).

| **Bioturbation potential of *H. scabra*$^{-1}$** | | |
|---|---|---|
| Defecation rate | 26 sediment pellets h$^{-1}$ × 10 h d$^{-1}$ | 260 pellets d$^{-1}$ |
| Defecation quantity | 260 pellets d$^{-1}$ × 0.36 g pellet$^{-1}$ | 93.6 g d$^{-1}$ |
| Annual defecation rate | 93.6 g d$^{-1}$ × 365 d | 34.164 kg ind y$^{-1}$ |
| **_H. scabra_ population bioturbation (1,000 m$^{-2}$)** | | |
| *ca.* 310 *H. scabra* | 310 ind. × 34.164 kg ind y$^{-1}$ | 10,590.84 kg y$^{-1}$ |
| **Available sediment (upper 5 mm)** | | |
| Area | | 1,000 m$^2$ |
| Volume of sediment | 1,000 m$^2$ × 0.005 m | 5 m$^2$ |
| Weight of sediment | 5 m$^2$ × 0.83 g cm$^{-3}$ | 4,150 kg |
| **Sediment turnover rate (upper 5 mm 1,000 m$^{-2}$)** | | |
| | 10,590.84 kg y$^{-1}$/4,150 kg | 2.55 times y$^{-1}$ |

treatment areas at $T = 1$ and $T = 3$ (Mann–Whitney $U$-test; $p < 0.01$, at both $T = 1$ and $T = 3$). Apart from an initially higher SOC rate in the *natural* compared with the *uncaged control* treatment at $T = 0$ (Mann–Whitney $U$-test; $p < 0.01$), there were no differences throughout the experimental period. *Uncaged control* increased significantly from $T = 0$ to $T = 1$ (Mann–Whitney $U$-test, $p < 0.01$), while *natural* did not over this time period (Mann–Whitney $U$-test, $p > 0.05$). *Natural* and *high* treatments only showed a significant difference (Mann–Whitney $U$-test, $p < 0.01$) at $T = 3$, while *natural* and *exclusion* treatments only exhibited significant differences at $T = 0$ and $T = 1$ (Mann–Whitney $U$-test, $p = 0.02$ and $p < 0.01$, respectively).

## Sediment turnover

On average each animal produced 26 sediment pellets h$^{-1}$ ± 2, which had a mass of 0.36 g ± 0.01. Observations of *H. scabra* at the study site showed individuals of this species actively fed for 10 h day$^{-1}$ during daylight hours (0600–1800 h) and were inactive for 14 h day$^{-1}$; *H. scabra* at the site generally remained buried during low tide. During daylight hours *H. scabra* were found at a density of 31 ind. 100 m$^{-2}$; this equates to 372,000 *H. scabra* on Natuvu's ca. 1.2 km$^2$ reef flat (S Lee, 2015, unpublished data). Bulk density of sediment at the site was 0.83 g cm$^{-3}$ ± 0.01. After incorporating these findings, calculations (Table 1) showed that the *H. scabra* population on Natuvu's reef flat had the potential to rework 10,590.84 kg sediment dry weight 1,000 m$^{-2}$ y$^{-1}$. Given the bulk density of sediment at the site, the upper 5 mm of sediment in an area of 1,000 m$^2$ would weigh 4,150 kg. Assuming sea cucumbers consume sediment to a depth of 5 mm, which is based on observations noted in *Uthicke (1999)*, the estimated population of *H. scabra* in 1,000 m$^2$ has the potential to rework the upper 5 mm of sediment approximately two and a half times a year.

## DISCUSSION

The current study explored the impact of holothurian removal on sedimentary function in a unique *in situ* setting. Despite the considerable 'background noise' from inherent and uncontrollable factors such as waves, wind, currents and marine benthos, several parameters provided clear evidence that overharvesting of sea cucumbers has a potentially strong effect on the ability of sediment to function as a biocatalytic filter system. The estimated rates of sediment turnover observed here for *Holothuria scabra* alone are substantial, and are more than twice the turnover recorded for a mixed population of *H. atra* and *Stichopus chlorontus* (4,600 kg dry wt $yr^{-1}$ 1,000 $m^{-2}$), which was estimated to be equivalent to turning over the upper 5 mm of sediment in that area annually (*Uthicke, 1999*).

Grain size selectivity has not been conclusively demonstrated for *H. scabra* (e.g., *Wiedmeyer, 1982*; *Tsiresy, Pascal & Plotieau, 2011*). However, within the current study, changes to grain size composition in areas of high *H. scabra* densities and the lack of any changes in their absence indicate that *H. scabra* are selective to certain grain sizes. This suggests that when sea cucumbers are present on inshore reef flats in high densities, they play a key role in the physical reworking and change of sediment structure in marine ecosystems. Based on the observed non-significant trends, a higher number of samples are suggested for future work. Several sea cucumber species are able to change sediment grain size through dissolution of calcium carbonate via acidity, and potentially abrasion processes, in their gut (*Hammond, 1981*; *Schneider et al., 2011*).

Throughout the study SOC rates and OPD exhibited a 'buffered' response in *high* treatment enclosures compared to *exclusion* enclosures, where they exhibited several large fluctuations. This buffered response was characterized by a predictable pattern with relatively low amplitude changes. Higher SOC rates in the *exclusion* treatment were consistent with findings by *Nickell et al. (2003)*; these higher SOC rates were likely driven by the increased respiration of benthic fauna, microbially-mediated oxygenation of OM, and reduced inorganic metabolites (*Kristensen, 2000*; *Nickell et al., 2003*). Deviations from this trend occurred at $T = 2$ and $T = 4$, following coastal flooding ten days prior to $T = 2$, and storm surge and coastal flooding five days prior to $T = 4$. Differences between the $T = 0$ measurements and the strong increase in the *uncaged control* between $T = 0$ and $T = 1$, compared to an absence of changes in the *natural* treatment over the same time, could be attributed to sampling during different tides, which was rectified for later time-points. Sea cucumbers actively feed on OM, thus reducing its concentration within the sediment (e.g., *Uthicke & Karez, 1999*; *Wolkenhauer et al., 2010*). Thus, there should be a higher OM concentration in the sediment of the *exclusion* treatment compared with that of the *high* treatment. Relatively higher OM concentrations facilitate increased microbial abundance and activity (*MacTavish et al., 2012*), in line with the observed higher SOC rates in *exclusion* treatment enclosures. Similarly, coastal flooding (observed at the study site ten days prior to $T = 2$ and five days prior to $T = 4$) likely delivered a strong OM-'pulse' onto the reef flat (*Briand et al., 2015*), explaining the elevated SOC rates even in *high* treatment enclosures following flooding (Fig. 2C).

The buffering capacity provided by high sea cucumber densities is likely due to the effect of the animals' feeding, excretion, and bioturbation. Feeding by sea cucumbers acts to 'clean' sediments, whereas the subsequent digestion and excretion of waste products plays an important role in nutrient recycling (*Purcell et al., 2016*). The recycled nutrients stimulate the growth of benthic microalgal communities (e.g., *Uthicke & Klumpp, 1998*; *Uthicke, 2001a*), which absorb nutrients and produce $O_2$ through photosynthesis (*Stockenreiter et al., 2016*). Bioturbation increases advective flow into and within the sediment directly through burying, burrowing, and bio-irrigation (*Meysman, Middelburg & Heip, 2006*), and indirectly as it increases bed sediment complexity (bioroughness). Bioroughness creates pressure gradients that can increase advective porewater flow up to seven-fold compared to smooth-bed flows (*Thibodeaux & Boyle, 1987*; *Huettel & Gust, 1992*), increasing the supply of degradable material and electron acceptors (such as $O_2$) to the sediment (*Rusch et al., 2006*), and thus promoting aerobic degradation. As aerobic degradation of OM in marine sediments is *ca*. ten times faster than anaerobic degradation (*Kristensen, Ahmed & Devol, 1995*; *Hulthe, Hulth & Hall, 1998*), sea cucumbers promote the efficient degradation of OM in a closely coupled benthic recycling system.

As grain size distribution showed no significant difference between *high* and *exclusion* treatment sediments at $T = 5$, simultaneous changes of OPD are assumed to be caused by the presence of sea cucumbers and not an alteration in sediment structure following storm surge prior to $T = 4$. Exceptionally high temperatures were recorded at $T = 5$ (31.2 °C ±0.07) compared with earlier months ($T = 0$ to $T = 3$; 25.74 °C ± 0.16) as a result of the 2015/2016 El Niño (*Blunden, 2017*). High water temperatures recorded at $T = 5$ coincided with the substantial reduction in sediment OPD of *exclusion* treatment enclosures. Warmer water temperatures induce microbial growth and respiration (*Nydahl, Panigrahi & Wikner, 2013*), and the relatively calm conditions for the same time period resulted in limited wave action and thus limited mixing of bed sediment and the overlying water column. Both these factors contribute to an increased SOC and a reduced OPD (*Kristensen, 2000*; *Friedrich et al., 2014*). The *high, natural* and *uncaged control* treatments did not experience a rapid reduction in OPD, despite being subject to the same environmental conditions. The abrupt and significant reduction in OPD during $T = 5$ in the *exclusion* treatment was likely caused by the combined manifestation of local and global stressors: the removal of sea cucumbers and the coinciding rapid increase in sea surface temperature.

Storm surge five days prior to $T = 4$ had visibly shifted and re-suspended a considerable amount of sediment at the study site. This disturbance would have functioned similarly to bioturbation (e.g., by oxygenating sediment and redistributing OM), explaining the significantly decreased SOC rate in *exclusion* enclosures at $T = 4$ (Fig. 2C). Disturbances such as storms, which are projected to increase in intensity as a consequence of climate change (*Emanuel, 2013*; *IPCC, 2014*), may thus be able to induce sufficient mixing of bed sediment capable of reducing SOC. However, the same models predict an increase in the severity, frequency, and duration of extreme precipitation events and heat waves (*IPCC, 2014*). While the global stressor of increased sea surface temperature in the current study had an effect on sediment function in all treatments, the high densities of sea cucumbers in the *high* treatment were able to buffer such extreme changes, particularly to OPD.

These results suggest a synergistic interaction between local and global stressors, whereby reductions in bioturbation caused by local overexploitation of a key benthic species (*Solan et al., 2004*) in combination with elevated temperatures led to an amplified effect. As mean sea surface temperatures, warm temperature extremes, and heavy precipitation events are forecast to increase and occur at higher frequency and intensity (*IPCC, 2014*), bioturbation becomes increasingly important to maintaining and improving sediment quality. Given the mechanisms of benthic-pelagic coupling (*Wild, Tollrian & Huettel, 2004*), the increased SOC rate and reduced OPD observed in areas void of *H. scabra* could contribute to deoxygenation of the overlying water column. Hypoxia affects the distribution, abundance, diversity and physiological state of benthic communities (*Meyer-Reil & Köster, 2000*), and increases the risk of hypoxia-derived mortality for larval stages of several marine organisms (*Vaquer-Sunyer & Duarte, 2008*).

High densities (350 g m$^{-2}$) of the sea cucumber *H. scabra* seem able to enhance the buffering capacity of sediment to stressors such as increased OM load and anoxia. This finding may have particularly strong implications in shallow restricted bodies of water, which are prone to large temperature variability, such as that of Marovo Lagoon, Solomon Islands. In June 2011, the lagoon experienced a large-scale harmful algal bloom, and its subsequent senescence resulted in large-scale hypoxia-derived mortality (*Albert, Dunbabin & Skinner, 2012*). The authors of the report suggest this event was caused in part by an increase in the catchment nutrient pool, and reduced processing of sediment nutrients and oxygenation of sediment following 15 years of extensive overharvest of sea cucumbers, combined with prolonged warm and calm weather conditions (*Albert et al., 2011*). As the frequency and severity of record high global temperatures and El Niño events are projected to increase with climate change (*Cai et al., 2014*; *Blunden, 2017*), and coastal waters are becoming increasingly eutrophic (*Fabricius, 2005*), events such as that reported for Marovo Lagoon may become more frequent. As such, the role sea cucumbers play in the bioturbation of inshore sediment becomes increasingly essential to the buffering capacity of tropical coastal marine ecosystems.

## CONCLUSION

This study builds upon previous work investigating the ecosystem role of sea cucumbers. Sandfish (*H. scabra*) are shown to play a demonstrable function in the role of sediment as a biocatalytic filter system. The combination of global stressors such as elevated sea surface temperatures and the local stressor of sea cucumber removal is of particular concern, as our results suggest potential synergisms resulting in an amplified effect. A consequence of the extensive overexploitation of sea cucumbers is that ecosystem functions and services offered by coastal marine environments in which these animals have been removed is likely compromised. These changes will leave coastal human communities increasingly vulnerable, particularly those of small Pacific islands that rely heavily upon such ecosystem services for their livelihoods and food security. We recommend that management should maintain moderate (in this study 60 g m$^{-2}$) to high densities (350 g m$^{-2}$) of sea cucumbers within reef ecosystems. Such densities should allow sea cucumbers to fulfil their role in maintaining sediment function at a scale that has a measurable impact.

## ACKNOWLEDGEMENTS

We thank the Leibniz Centre for Tropical Marine Research (ZMT), Wildlife Conservation Society, University of Bremen, and University of the South Pacific for supporting field and laboratory work. We would also like to thank Dr. Steven Purcell for comments that helped to improve this manuscript. Finally, vinaka vakalevu to the people of Natuvu village for allowing this study within their traditional fishing ground. Parts of this work have been published in a non-peer reviewed report (*Lee et al., 2017*) and as an MSc thesis (*Lee, 2016*).

### Funding

This study forms part of the REPICORE project (Resilience of Pacific Island coral reef social-ecological systems in times of global change) funded by the (German) Federal Ministry of Education and Research (BMBF) through the program 'Nachwuchsgruppen Globaler Wandel 4 + 1' (grant number 01LN1303A). The work was further supported by the Rufford Small Grants Foundation (RSG 17605). The research reported in this paper contributes to the Programme on Ecosystem Change and Society (http://www.pecs-science.org). The funders had no role in study design, data collection and analysis, decision to publish, or preparation of the manuscript.

### Grant Disclosures

The following grant information was disclosed by the authors:
German Federal Ministry of Education and Research: 01LN1303A.
Rufford Small Grants Foundation: RSG 17605.

### Competing Interests

The authors declare there are no competing interests.

### Author Contributions

- Steven Lee conceived and designed the experiments, performed the experiments, analyzed the data, contributed reagents/materials/analysis tools, prepared figures and/or tables, authored or reviewed drafts of the paper, approved the final draft.
- Amanda K. Ford conceived and designed the experiments, performed the experiments, contributed reagents/materials/analysis tools, authored or reviewed drafts of the paper, approved the final draft.
- Sangeeta Mangubhai contributed reagents/materials/analysis tools, authored or reviewed drafts of the paper, approved the final draft.
- Christian Wild conceived and designed the experiments, contributed reagents/materials/analysis tools, authored or reviewed drafts of the paper, approved the final draft.
- Sebastian C.A. Ferse conceived and designed the experiments, contributed reagents/materials/analysis tools, prepared figures and/or tables, authored or reviewed drafts of the paper, approved the final draft.

## Data Availability

Lee, Steven; Ford, Amanda; Mangubai, Sangeeta; Wild, Christian; Ferse, Sebastian CA (2017): Effects of sandfish (*Holothuria scabra*) removal on shallow-water sediments in Fijiby sea. PANGAEA, https://doi.org/10.1594/PANGAEA.883604, Supplement to: Lee, S et al. (submitted): Effects of sandfish (*Holothuria scabra*) removal on shallow-water sediments in Fiji. PeerJ

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
