# Peer review of "Effects of sandfish (Holothuria scabra) removal on shallow-water sediments in Fiji"

_PeerJ, doi:10.7717/peerj.4773_

## Round 0.1 · original submission · Minor Revisions

Please pay particular attention to shortening the ms where possible, making clear what data has been previously published and addressing the statistical questions posed by the reviewers.

Reviewer 1 ·

Basic reporting

The paper is generally well written. The Introduction provides a clear background to the study and the knowledge gap it addresses. The negative impacts of sea cucumber removal on marine ecosystems are often cited but there is little research to support or identify the mechanisms by which this occurs. This work contributes to our body of knowledge in this area.
Literature was well referenced and relevant, although the reference list was very long for a paper this size.
Lines 93-95: Not all sea cucumbers play an important role in the recycling and remineralization of organic matter or are endangered. Authors should note that it is predominantly the deposit feeding, Aspidochirotid (and a few Dendrochirotid species) sea cucumbers that are commercially exploited and perform this ecological role.
Lines 93-113: There is excessive detail in this paragraph on the issue of stock decline (particularly lines 99-105). It is good to make the point and reference it but so much specific information on individual countries is laboring the point. Please try to reduce the text here.
Lines 121-124: a key statement “The current study aimed to understand how the removal of sea cucumbers affects sedimentary function through in situ experimental manipulations of H. scabra densities, thus providing information on the poorly understood ecological implications of sea cucumber fisheries.” Is currently buried in the Introduction. The authors could consider including this statement in a small paragraph at the end of the Introduction to logically where they can clearly lay out the paper’s rationale, i.e. sea cucumbers are important, they have been overexploited, sandfish in particular are important and this is how we are going to demonstrate it. Paragraph lines 115-141 can focus on the more technical aspects of the study.
Lines 124-125: more suited to the Methods section.
Lines 135-136: “These parameters provide a proxy for the function of sediment as a biocatalytical filter system” would be better placed sooner in the text prior to discussion of the parameters. This would justify why you chose these particular variables to measure.
The referencing is very comprehensive (although as noted above, may be a bit too exhaustive in places, I leave it up to the authors to see if there could be some trimming back of references. Some comments on the references:
Holothuria scabra is not italicized in all references.
Line 498: Gilbert et al. (2014) is not referred to in the ms.
Line 327: Mezali & Soualili, 2013 in ms but not references.
Some references are in wrong order, e.g. Purcell (2004), Uthicke (1999), Huettel et al. (2006), Mercier et al. (1999).

Experimental design

The field design is sound and well designed and executed, often in difficult circumstances by the sound of it. Particularly notable was the use of controls for the fencing which would account for any cage effects that may have occurred. The methods are mostly well described, although could be tightened up and made more clear in parts. I have made some suggestions to this effect but they are not exhaustive.
The primary sediment sampling frequency is not described (despite mentioning twice that stocking density was checked 6 times a week). I had to glean from the Results (and a sentence in the Introduction) that there was monthly sampling with 6 sampling events between September 2015 and February 2016. Furthermore, August 2015 is included in Figure 1, even though Time 0 is September 2015 (Line 170). Please include a clear outline of the sampling schedule early on in the Results section. Thereafter, use T=0, T=1 consistently (including Figure 2).
Lines 175-176: ‘All animals used to restock enclosures were collected from the immediate area around each respective enclosure’ but the results do not mention any restocking. If loss or escape of sandfish was an issue, this should be noted and discussed or not mentioned in the methods.
OPD was only measured from Time 2 onwards (Line 279 in Results). This should be stated in the Methods with an explanation of why.
Lines 218-221: This requires some clarification. Where were the individuals for the sediment turnover experiment monitored? If this was outside the pens in the most common habitats, please clearly say so, also check the n, 17+6+4=27, not 28. Also, it is not clear whether 6 pellets were collected from every hourly sample or just once during the 4-hour faeces monitoring? What is the relevance of a polymerase chain reaction tube given they were then dried in an oven and weighed? Would ‘collected in a vial’ work as well?
The method of analysis is not clearly explained. Also, I question the separate analysis of the two controls (natural vs control) together and the density treatments (high vs exclusion) together. The authors have been prudent in comparing natural vs control to determine whether cage effects occurred before proceeding with the density differences, this is well done. However, there has been a lost opportunity to compare exclusion, natural (i.e. low) and high densities which would have provided further fine-scale information on what density of sandfish might provide the ecosystem benefits that are being investigated. In fact, the abstract implies that this what was done (Lines 34-36). The data analysis is complicated because the storm events interrupt what may have occurred if the treatments had remained in place for the entire 6 months. This needs to be dealt with in the most clear, simple and understandable way. After establishing that there were no cage effects, the control treatment could be dropped and the three density treatments (exclusion, natural or low, and high) analysed. It may help clarify the results later.
Results are only given for Chi-squared analyses and Mann Whitney-Wilcox U Tests. No repeated measures ANOVA results are presented.

Validity of the findings

Overall, the findings are valid and some interesting results emerge from the study. However, the variability in the results and the effects of the two disturbances suggest that more caution is needed in the interpretation (refer also to general comments).
Results on Grain size distribution (Lines 267-276) are slightly confusing and fragmented. This is obviously tricky due to the storm events but the authors could consider moving Lines 272-276 to the start of the paragraph as these tests were done first. The main result for high treatment could follow with emphasis that this is the only significant result (if that is the case). A sentence on the effect of the storm surge between T3 and T4 (if anything) would be informative.
As mentioned above, Line 279 ‘OPD measurements began at T=2 of the experiment’ should be stated earlier in the Methods. If the timing of the storm surge is relevant to this result, please provide more results here on what happened during this period.
I’m afraid I found the OPD results very difficult to reconcile with Figure 2b. The difficulty of differentiating the bars was the main obstacle. I can’t be sure if there is a problem but I would request the authors to carefully check the stated values against the graph and to make sure this is easy to read in a published version.
Results on SOC (Lines 291-299) are also confusing. From Figure 2c, it also appears that cage control SOC also increased substantially from T0-T1, while natural density remained consistent but these are not mentioned. The graph does not seem to support the statement ‘Sediment in exclusion treatment areas generally had a higher O2 uptake than high treatment areas over the course of the study’ (Lines 294-295), except for T1 and T3 and the latter standout value is not mentioned, except in the buffering context. Please keep the terminology consistent (O2 uptake or SOC).
Results on Sediment turnover (Lines 302-311) would be improved if the main result (as dictated by the Methods) is presented first, followed by the supplementary information which allowed estimation of the turnover rate. e.g.
‘On average each H. scabra individual produced 26 sediment pellets h-1 ± 2, which had a mass of 0.36 g ± 0.01. Observations of H. scabra at the study site showed this species actively fed for 10 h day-1 during daylight hours (0600-1800 hrs), and generally remained buried during low tide. During daylight hours H. scabra were found at a density of 31 ind. 100 m-2. Bulk density of sediment at the site was 0.83 g cm-3 ± 0.01. After incorporating…’
This sediment turnover figure is pretty impressive but Line 310 should state that ‘…the estimated population in 1000 m2 at the study site has the potential….’
The abstract includes a statement on turnover that is not included in the ms. ‘Findings revealed that the natural population at the study site can rework ca. 10590 kg dry sediment 1000 m-2 year-1; more than three times the turnover recorded for H. atra and Stichopus chloronotus’. Please check that the entire abstract accurately reflects the ms.
Daylight hours are defined as 0600-1800 hrs in this study. Many studies indicate that sandfish are more active in the afternoon and evening (e.g. Wolkenhauer, 2008. Burying and feeding activity of adult Holothuria scabra (Echinodermata: Holothuroidea) in a controlled environment. SPC Beche-de-mer Information Bulletin, 27, 25-28). The authors state that sandfish numbers were determined by a pilot survey at the study site but should specify whether the survey was carried out at a suitable tide and time of day to optimize counts and hence provide confidence in the population estimates. Or explain if they are sure that sandfish at this site did not exhibit any greater activity at particular times of day.
I found Figure 2 very hard to understand and it must be improved. Suggested Figure 2 modifications:
• Use T0, T1, T2, etc, not dates as you refer more to times than date in the ms.
• It would be useful to show significant differences on Figure 2, where relevant.
• Caption should read something like ‘Plots of results for temperature and rainfall (a), oxygen penetration depth (b), and sediment oxygen consumption (c), at each sampling time’.
• It might be helpful to note the time of the unusual storm events on the graphs with an arrow or similar.
• Does the legend in 2b match the order of the bars in the graph? The grey shades don’t seem to.
• I assume the OPD is presented downwards to represent the direction that this is measured in the sediment but it does not help interpretation of the graph because when we are discussing high and low values we envision the opposite. This is my opinion only, leave it to the authors to decide.
More on Figure 2b. The methods state that cage control and natural treatments were ‘controls’ and analyzed together, as were the exclusion and high treatments. The graph would be more easily interpreted if the two sets were placed side by side in graph 2b. Also note that the bars are insufficiently distinguishable, different shades should be used or dots, hatched bars, etc. Similarly for graph 2c, possibly use same shapes open and closed for the paired analysis sets of data? Graph 2c in particular is very noisy and hard to interpret and I found difficult to reconcile with the Results description in the text. If they can’t be shown more clearly, graph them separately. Alternatively, remove the control treatment if they do not show any difference and are obscuring the results you are trying to highlight.
Line 137 in the Introduction “We hypothesized that high densities of H. scabra would facilitate the efficient degradation of OM.” Why didn’t you measure OM? This is one of the most basic parameters considered in virtually every study of sea cucumbers and their use of and interaction with sediment. There are also suggestions of higher OM concentration in the sediment of the exclusion treatment (Lines344-346), that flooding likely delivered a strong OM-pulse to the reef (Lines 348-349) and so on. Many conclusions are drawn from these ‘potential’ effects which cannot be proven.
Discussion comments:
Line 326 should refer to ‘several sea cucumber species’.
This paragraph should include additional references below noting the complexity of grain size selectivity wrt sandfish. None of the species in the references cited are H. scabra, so sediment selectivity is being assumed. Results would be better described as providing possible indications of grain size selectivity or cautious support to the presence of grain size selectivity. Wiedmeyer (1982) concluded that sandfish select for smaller particle size, especially smaller individuals. However, Tsiresy et al. (2011) did not find evidence for selectivity below 2 mm grain size. He discusses sediment compactness as a factor to be considered in sandfish growth and thus in how they interact with the sediment, potentially affecting the results presented in the ms (for example, storm surges which redistribute sediment may affect compactness even if grain size is unaltered).
Tsiresy et al. (2011) An assessment of Holothuria scabra growth in marine micro-farms in southwestern Madagascar. SPC Beche-de-mer Information Bulletin, 31, 17-22
Wiedmeyer (1982) Feeding behavior of two tropical holothurians Holothuria (Metriatyla) scabra (Jager 1833) and H. (Halodeima) atra (Jager 1833), from Okinawa, Japan. Proceedings of the 7th International Coral Reef Symposium, Guam, 2, 853-860.
Conclusions: lines 448-449, possibly more so for Pacific and other island communities where livelihoods and food security are heavily reliant on these services?
The second half of the Discussion should be reduced and tightened up (from Lines 398-440). The consequences of a malfunctioning sediment biocatalytical filter system in habitats under stress, synergistic effects and how sandfish may or are likely to influence these functions are important. Lines 435-440 are repetitive of, and should be incorporated into, the Conclusion.

Additional comments

The paper is valuable because it involves a high-value commercial species that is believed to be beneficial to the marine environment and has been decimated in most areas where it occurs. The field work has been well done, using novel parameters, and will be of interest to field workers and researchers. I have made some suggestions to make the Methods more readable and clearer. The Results were variable and were affected by other factors which further increased noisiness: a minor sampling issue with the T0 baseline sampling, then storm events affected 2 out of 5 samples, therefore nearly half of the sample times have no significant differences between variables (this is how it appears on the graphs although the written results are presented in a rather piecemeal way). Heavy rain and storms may have had other impacts not considered in the paper and not detected by the analyses, for example, due to freshwater inundation, changes in sediment compactness, etc. The authors mention the numerous factors that they couldn’t control at the start of the Discussion and handle the results of the storm events quite well. There are some good results and they are worth reporting and emphasizing. However, I felt the authors took the sediment function processes scenario too far on fairly limited data. The speculative tone used in lines 418-433 is reasonable but Lines 435-437 draw too strong a conclusion from a six-month study considering the issues noted above. The Discussion should be revised to concentrate on the parameters measured and adopt more cautious conclusions.

Reviewer 2 ·

Basic reporting

Abstract:
You say “this fishery” but there are more than 80 fisheries globally. Which one do you refer to?
Are these all “sedimentary functions”? Consider rewording.
If the “natural population at the study site can rework ca. 10590 kg dry sediment 1000 m-2 year-1”, does this mean the population at that site can rework many of these units? The choice of units looks strange. Is there a way to express in kg reworking per animal per year?

Introduction:
Is “biocatalytical” even a word?

Maybe at line 80 put a paragraph break

Line 132: can you add more detail to elaborate on HOW these animals can “alter grain size distribution”?

Experimental design

Line 159: “plots without mesh” doesn’t sound like a cage control. I encourage the authors to check up on some of the earlier work (decades ago) about cage controls. Rewording is needed here, or extra clarification.

Line 183-184 says that the sediments were placed into containers then dried. Normally for this sort of work, they would be rinsed with freshwater before drying. How might salt have biased the results? Please mention.

Line 225: surely the sediment defecation rate depends on the time of day. Give the range in time that this sampling was done, and give some citations about whether time of day might bias defecation rates. Also, how are these values scaled up to one year, given that this species apparently can bury in sediments for some parts of the day and might be relatively inactive in the cooler time of year? More consideration needed here.

Line 226: do sea cucumbers have a “caudal end” or is it just a posterior end? Caudal sounds like a fish term.

Line 228: how were length and weight recorded? Give detailed methodology please.

Analyses:

Is Chi-squared test the industry standard for comparing grain size distributions? I believe that this analysis would treat each grain size class as an independent group to compare against the others, which they are not. How did you account for the non-independence of adjacent size classes? A better approach is to use an analysis more suited to the log-normal distribution, and non-independent aspect of how these data are structured. The geological community is quite adept at the proper approach. The data probably need to be reanalysed with a more suitable test.

Validity of the findings

Line 307 states that the “H. scabra population on Natuvu’s reef flat”. But how big is this reef flat and how big is the population?

Line 309 says “Assuming sea cucumbers consume sediment to a depth of 5 mm”. This doesn’t sound very plausible. What have other studies shown? As these are deposit feeders, with peltate tentacles, it is probably more like the first 1-2 mm. Please check and add more references here to back up this statement, as it is not convincing to me.

Additional comments

In running a routine authenticity check, I discovered that part of this study had been published in a couple sources already (one as an abstract in a published report, and the other as a chapter in a published report). Can the authors affirm that the data and results in this manuscript have not been published elsewhere? Also, it might be important to cite the other publications in which these findings or data have also been presented.

---

## Round 0.2 · Minor Revisions

Dear Steven, thanks for your readiness to address the suggested changes, but there are a couple of minor corrections still required. Your new insertions have on one occasion used differing abbreviations (hr) can you correct please. I also noted at least one occasion in which species names are not italicised , can you also check. Otherwise, happy to accept.

---

## Round 0.3 · accepted · Accept

Thank you for attending to review comments promptly and efficiently.

#